# Persistence Homology Distillation for Semi-supervised Continual Learning

**Yan Fan, Yu Wang**\*, **Pengfei Zhu, Dongyue Chen, Qinghua Hu**
College of Intelligence and Computing, Tianjin University, China
Haihe Laboratory of Information Technology Application Innovation, China
`fyan_0411@tju.edu.cn, wang.yu@tju.edu.cn, zhupengfei@tju.edu.cn`
`dyuechen@tju.edu.cn,huqinghua@tju.edu.cn`

## Abstract

Semi-supervised continual learning (SSCL) has attracted significant attention for addressing catastrophic forgetting in semi-supervised data. Knowledge distillation, which leverages data representation and pair-wise similarity, has shown significant potential in preserving information in SSCL. However, traditional distillation strategies often fail in unlabeled data with inaccurate or noisy information, limiting their efficiency in feature spaces undergoing substantial changes during continual learning. To address these limitations, we propose Persistence Homology Distillation (PsHD) to preserve intrinsic structural information that is insensitive to noise in semi-supervised continual learning. First, we capture the structural features using persistence homology by homological evolution across different scales in vision data, where the multi-scale characteristic established its stability under noise interference. Next, we propose a persistence homology distillation loss in SSCL and design an acceleration algorithm to reduce the computational cost of persistence homology in our module. Furthermore, we demonstrate the superior stability of PsHD compared to sample representation and pair-wise similarity distillation methods theoretically and experimentally. Finally, experimental results on three widely used datasets validate that the new PsHD outperforms state-of-the-art with 3.9% improvements on average, and also achieves 1.5% improvements while reducing 60% memory buffer size, highlighting the potential of utilizing unlabeled data in SSCL. Our code is available: https://github.com/fanyan0411/PsHD.

## 1  Introduction

Continual learning involves developing practical approaches for incrementally training models, enabling them to learn new concepts while effectively retaining previously acquired knowledge [1]. The problem of catastrophic forgetting in continual learning impacts the model's performance on previously learned tasks. Numerous efforts have been dedicated to addressing this issue in continual learning [2–8]. However, most state-of-the-art methods are trained under the assumption of fully labeled data, neglecting the vast amounts of unlabeled data.

Semi-supervised continual learning introduces unlabeled data in CL by partially annotating the trained data in each task and aims to alleviate catastrophic forgetting by exploiting these unlabeled data[9–14]. Learning from unlabeled data to improve model generalizability across tasks is a natural approach, but its performance significantly relies on the size of the memory buffer [12, 15]. Generative replay methods reduce the need for a memory buffer but incur prohibitive costs for higher resolution images and are limited to relatively simple datasets [11, 12]. Knowledge distillation methods are proposed to preserve valuable information in the unlabeled data of the previous model [13, 14].

---

\*Corresponding author

38th Conference on Neural Information Processing Systems (NeurIPS 2024).

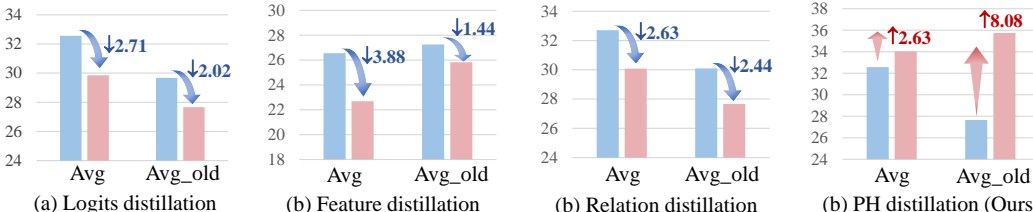

Figure 1: The performance interference of extra unlabeled data distillation. The blue columns represent experiments only disillation knowledge only on labeled data, and the red columns are results with extra unlabeled data distillation based on blue ones. The four approaches are (a) iCaRL, (b) PodNet, (c) R-DFCIL and (d) our persistence homology distillation method. Avg and Avg_old mean the average incremental accuracy of all tasks and old tasks, respectively.

Logits or feature distillation approaches focus on sample-wise representation [9]. Relation distillation methods, as variants of representation distillation, are explored for vulnerable representation of unlabeled data to enhance more expressive knowledge preservation, including class-instance similarity [13], and local structural similarity [14]. Although these distillation strategies have proven effective, the impact of noisy relations remains a significant challenge. As shown in Figure 1, the succeeded biased or inaccurate information often interferes with the generalization on known classes impact and even impact the entire tasks[16]. Therefore, we argue that a more powerful distillation that investigates noise insensitive information preservation in unlabeled data is needed for semi-supervised learning continual learning.

To address the aforementioned problem, we explore extracting multi-scale shape and structural information from unlabeled data that is insensitive to feature space bias and noise interference in semi-supervised continual learning. Inspired by the manifold assumption [17] and the homomorphism between manifolds and simplicial complexes [18], we explore summarizing complex vision data into simplicial complexes, and investigate the preservation of multi-scale structural properties despite representation changes. Persistence homology in topological data analysis, as an adaptation of homology to discrete metric spaces, reflects the homology-class evolution across different scales. Its characteristics of invariants on homeomorphism and robustness to perturbations provide an adaptable view to explore the structure in unlabeled data.

To this end, we propose a new model named Persistence Homology Distillation (PsHD) for semi-supervised continual learning to stabilize learned tasks by leveraging the power of topological features. Firstly, we construct local simplicial complexes in vision data based on the weighted k-hop neighborhood and extract multi-scale structural features using persistent homology. Furthermore, we devise a model parameter-independent distillation loss based on these structural features and also design an acceleration algorithm to mitigate the computation costs. We also demonstrate the stability privilege of persistence homology distillation compared to pair-wise similarity distillation strategies [13, 19, 20] theoretically and experimentally. The primary contributions of our work are as follows:

- We utilize simplicial complexes to approximate vision data and explore stable topological feature representation of unlabeled data in semi-supervised learning.

- We propose a novel persistence homology distillation strategy for SSCL that is insensitive to noise information interference based on the intrinsic topological features, and devise an accelerating algorithm to reduce computation costs.

- We demonstrate that our method outperforms existing methods on several benchmarks, and highlights the potential of utilizing unlabeled data to overcome catastrophic forgetting.

## 2  Related Work

**Persistence Homology**    Topological data analysis (TDA) is a new approach to seeking structural or geometric information from discrete data. Persistent homology (PH) is one of the most popular tools

in TDA to capture topological information about connectivity and holes in geometric data. Several impressive applications of persistent homology in complex data are explored [21–25].

Topological features in vision data have been investigated broadly, including leveraging the prior topology knowledge to improve the segmentation results [26, 27] and exploiting high-order persistence spectral representations in image embedding [27]. These works are all based on explicit object-level topology features. Few insights about TDA in category-level vision data have been investigated. Liu et al. [28] propose a topological prediction model in image dataset, while is not learnable to meet the requirement of deep learning. Madhu and Chepuri [29] introduce the simplicial contrastive loss to preserve more expressive consistency, while only considering neighbors' aggregated representation. So far, there are limited results on learnable intrinsic category features to summarize vision data and investigate the structural stabilization of data distribution.

**Continual Learning** Continual learning methods can be categorized into three groups. The regularization-based methods selectively regularizing the variation of essential network parameters [2, 3] or preserve the outputs consistency of prediction functions [30, 31]. Replay-based methods store few representative samples for retraining or old knowledge distillation [4–6], or utilize an additional generative model to replay previous data [32, 33]. The parameter isolation methods isolate parameter subspaces dedicated to each task throughout the network[7, 8].

To address the limitation of continual learning in neglecting tremendous unlabeled data, recent efforts have been directed towards semi-supervised continual learning [9, 12, 11, 15, 13]. Some of these methods leverage the generalizability of unlabeled data to alleviate forgetting by incorporating gradients of unlabeled data [12] or employing contrastive loss among unlabeled data [15]. Such strategies require a substantial memory buffer to ensure satisfactory performance. To deal with the memory limitation, a generative reply method regularizes discriminator consistency to mitigate catastrophic forgetting [11]. Employing meta distribution to consolidate the task-specific knowledge [34] and training an adversarial auto-encoder to reconstruct images [35] extends [11] further. However, these methods incur prohibitive costs on higher-resolution images. Knowledge distillation strategies have also been explored on unlabeled data to address the forgetting in SSCL, including sample-wise representation [9], class-instance relationships [13], and local-neighborhood similarity [14].

In continual learning scenarios with limited labeled data, the required changes of feature space can often be dramatic [1], and the learned class-related representation is usually biased to labeled data [13]. With the unreliable distribution, stabilizing the feature extractor may interfere with accommodating new representations and subsequent task refinement. A more stable representation learning robust to data transformation is required for the SSCL.

## 3 Methodology

### 3.1 Preliminaries on Persistence Homology

A *k-simplex* $\sigma$ is a k-dimensional polytope which is the convex hull of its k+1 vertexes. A single point can be regarded as a 0-simplex, and a line segment can be defined as a 1-simplex. A 2-simplex is a triangle, and a 3-simplex can be seen as a tetrahedron. The convex hull of any nonempty subset of the k+1 points that define a $k$-simplex is called a *face* of the simplex. A *simplicial complex* $\mathcal{K}$ is a set of simplex that satisfies the following conditions. (1) Any face of a simplex from $\mathcal{K}$ is also in $\mathcal{K}$. (2) The intersection of any two simplices $\sigma_1, \sigma_2 \in \mathcal{K}$ is a face of both $\sigma_1$ and $\sigma_2$. A common choice of simplicial complex $\mathcal{K}_r$ is the Vietoris-Rips simplicial complex $\mathcal{R}(X, r)$, in which a simplex $\sigma = \{x_1, x_2, \ldots, x_m\} \in \mathcal{R}(X, r)$ when $dist(x_i, x_j) < r$ for all $1 \leq i, j \leq m$.

A *filtration* $\{\mathcal{K}_r\}_{r \in \mathbb{R}}$ of a point cloud in $\mathbb{R}^d$ can be constructed from any filter function $f : \mathbb{R}^d \rightarrow \mathbb{R}$ by considering each simplicial complex to be the sublevel set of $f$ threshold by $r$, $\mathcal{K}_r = \{x \in \mathbb{R}^d | f(x) \leq r\}$. The underlying filter function introduced above is the distance function $\delta_X : \mathbb{R}^d \rightarrow \mathbb{R}$, where $\delta_X(y) = min\{\text{dist}(y, x) | x \in X\}$ is the distance from $y$ to point set $X$. To be specific, the Vietoris-Rips simplicial complex $\mathcal{R}(X, r)$ approximates the sublevel set

$$K_r = \delta_X^{-1}((-\infty, r]) = \{y \in \mathbb{R}^d | \delta_X(y) \leq r\} = \cup_{x \in X} B(x, r), \tag{1}$$

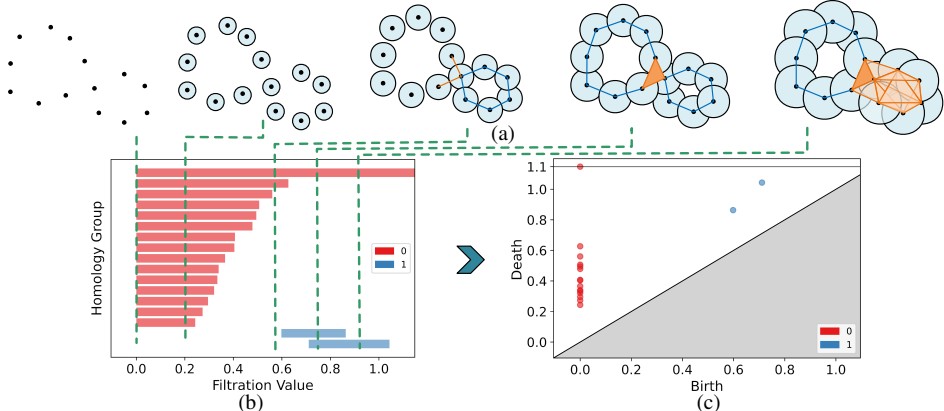

Figure 2: Illustration of topological data analysis. (a) Filtration of simplicial complex, (b) corresponding persistence barcode, and (c) persistence diagram.

where $B(x, r)$ is a ball with radius centered around $x \in X$. Then the filtration is a nested sequence of subcomplex set

$$\emptyset = K_0 \subset K_1 \subset \cdots \subset K_m = K, \tag{2}$$

which reflect multi-scale topological spaces underlying the point data, as shown in Figure 2(a). The persistence homology quantifies the changes in topological features as the subcomplexes grow with the increasing threshold of the filter function. The strict definition of persistence homology is provided in Appendix A.1. An easier way to understand is as follows. The lifetime of each multi-scale topological feature is recorded by persistence barcode, as shown in Figure 2(b). Finally, the persistence barcode can be converted into a persistence diagram, as shown in Figure 2(c). $Dgm = \{(x_\sigma, y_\sigma) \in \mathbb{R}^2 : x_\sigma < y_\sigma\} \cup \triangle$ are a scatter plot with the $x$ and $y$ axes respectively depicting the scale $x_\sigma$ at which each topological feature is born one subcomplex and dies in threshold $y_\sigma$ or is identified with another homology class. $\triangle$ is the diagonal set containing points in PH.

### 3.2 Persistence Homology Distillation for Semi-supervised Continual Learning

**Formulation of SSCL**  As shown in Figure 3(a), when encountering new data $D_t$, let $f_\theta^t$ be the model to be trained, with both labeled data $L$ and unlabeled data $U$ input into the model. The parameters $\theta$ are updated through the cross-entropy loss $\mathcal{L}_{CE}$ and semi-supervised loss $\mathcal{L}_{SSL}$ to adapt the model to the new task. We follow DSGD [14] to employ Fixmatch [36] as $\mathcal{L}_{SSL}$. Simultaneously, to prevent catastrophic forgetting, the memory buffer $M$ is used to regularize the knowledge consistency between $DK(h^t)$ and $DK(h^{t-1})$ based on the previous model $f_\theta^{t-1}$ through a representation distillation loss $\mathcal{L}_{rd}$. Now we propose a new persistence homology distillation loss for unlabeled data $\mathcal{L}_{hd}$ as part of the continual learning loss. The complete loss function for SSCL is as follows:

$$\mathcal{L}_{sscl} = \mathcal{L}_{CE} + \mathcal{L}_{SSL} + \mathcal{L}_{rd} + \lambda \mathcal{L}_{hd}. \tag{3}$$

**Persistence Homology Distillation**  Existing distillation methods preserve the sample representation, the class-instance similarity between tasks, and local neighbor similarity. This limits the performance when inaccurate representation is inherited or noise occurs in previous tasks, especially when a significant transfer between the feature spaces of two tasks exists. To exploit the intrinsic structural features preservation, which topological invariant regardless of specific representation, we propose a persistence homology-based distillation method and design an effective algorithm to achieve stable knowledge accumulation.

The first step of our model is to define the topological similarity among two samples. Given a collection of image sets, we build the local persistence diagram by considering the weighted k-hop neighborhood $\mathcal{N}(x_i, k)$. To be specific, let $G = (V, E)$ be the adjacency graph of memory buffer, where $V$ is the set of replayed image samples. $W \in \mathbb{R}^{N \times N}$ be the adjacency matrix of $G$ such that $W_{ij} = 1$ for $e_{ij} \in E$ and 0 for otherwise. As a discrete point data, we construct the adjacency matrix

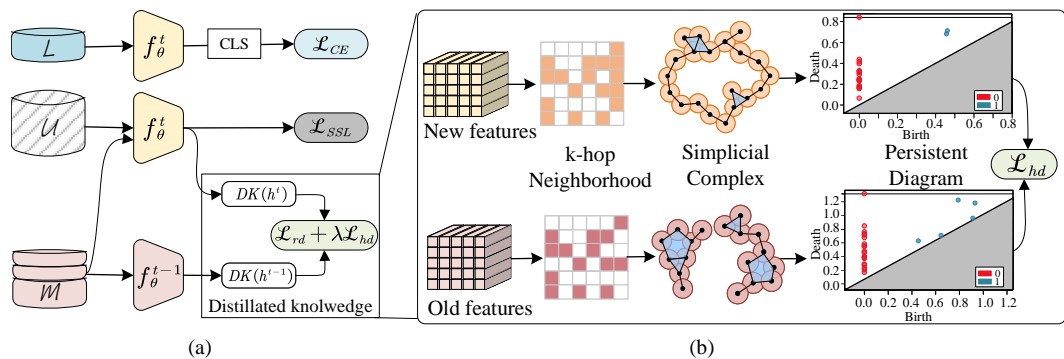

(a)                     (b)

Figure 3: Illustration of our proposed persistence homology distillation for semi-supervised continual learning. (a) represents the backbone of SSCL, $\mathcal{L}_{CE}$ and $\mathcal{L}_{SSL}$ are the cross-entropy loss on labeled data and semi-supervised loss on unlabeled data. $\mathcal{L}_{rd}$ means the representation distillation loss on the memory buffer. (b) corresponds to our PsHD loss employed on the replied unlabeled samples.

by applying a threshold to the cosine similarity $s_{ij} = <h_i, h_j> = \frac{h_i h_j}{\|h_i\|\|h_j\|}$

$$W_{ij} = \begin{cases} 1, & \text{if } s_{ij} \geq \beta \\ 0, & \text{if } s_{ij} < \beta, \end{cases} \tag{4}$$

where $h_i$, $h_j$ are the representations of sample $x_i$ and $x_j$, respectively. We use the logits $h_i \in \mathbb{R}^d$ in this study. Next, we can group the weighted k-hop neighborhood set $\mathcal{N}(x_i, k)$, where for any $x_j \in \mathcal{N}(x_i, k)$, the shortest path between $x_i$ and $x_j$ is at most $k$. Armed with the distance function $\text{dist}(x_i, x_j) = 1 - s_{ij}$, we can construct the h-dimensional persistence diagram $Dgm_h(\mathcal{N}(x_i, k))$ by considering the distance function as the filtration function according to Eq (1)-(2).

Subsequently, we try to preserve this structural information by designing a distillation loss, which goes beyond only the pairwise similarity of node features. Given any sample $x$ of the memory buffer, we can derive the persistence diagrams of the old task by the old feature extractor $Dgm_h(\mathcal{N}_x^k, f_{old})$. The corresponding persistence diagram in the new model is $Dgm_h(\mathcal{N}_x^k, f_{new})$, where the neighborhood set is the same as the previous task. The structural similarity between two tasks is measured with the p-Wasserstein distance between the corresponding persistence diagrams as

$$d_{\mathcal{N}_x^k}(f_{old}, f_{new}) = inf_\gamma \left( \sum_{u \in Dgm_h(\mathcal{N}_x^k, f_{old})} \|u - \gamma(u)\|^p \right)^{\frac{1}{p}}, \tag{5}$$

where $p \geq 1$ and $\gamma$ is taken over all bijective maps between $Dgm_h(\mathcal{N}_x^k, f_{old})$ and $Dgm_h(\mathcal{N}_x^k, f_{new})$. In our analysis p=1. This distance quantifies how much effort is needed to transform one topological feature representation into another.

**Persistence Homology Distillation Acceleration Algorithm** It is known that the worst-case complexity of generating a homology diagram is $\mathcal{O}(n^{2.5})$, where $n$ is the number of the simplices [37]. Thus, the complexity of computing local topology similarity for each sample in the memory buffer is $\mathcal{O}(|M|n^{2.5})$, which is computationally expensive. This procedure is also redundant since nearby samples typically share similar topological structures. To mitigate the computation cost, we design an approximate algorithm to simplify the final distillation loss and accelerate the persistence homology distillation procedure.

Given the replayed data $M$, we randomly select one sample $x_1$ and group its weighted k-hop neighborhood set $\mathcal{N}(x_1, k)$. We then select another sample $x_2$ in the remaining samples $M \backslash \mathcal{N}(x_1, k)$, and group its weighted k-hop neighborhood set $\mathcal{N}(x_2, k)$. This process is repeated until all samples are grouped into their respective neighborhood sets. After this step, we can compute several homology diagrams derived from grouped k-hop neighborhood sets $\{\mathcal{N}(x_1, k), \dots, \mathcal{N}(x_l, k)\}$. We design the final distillation loss as $L_{hd}$ in SSCL by meaning of all separated local homology diagrams

$$\mathcal{L}_{hd} = \frac{1}{|S|} \sum_{x \in S} d_{\mathcal{N}_x^k}(f_{old}, f_{new}), \tag{6}$$

where $|\cdot|$ is the cardinality. This procedure is summarized as Algorithm 1 in Appendix A.2. By doing so, the complexity is reduced from $\mathcal{O}(|M|n^{2.5})$ to $\mathcal{O}(|S|n^{2.5})$. We experimentally find that the size of $S$ is approximate to the number of classes in the memory buffer, so the $\mathcal{O}(|S|n^{2.5})$ is approximated to be $\mathcal{O}(C_M n^{2.5})$. This contributes to a significant reduction in computation, approximately 10-fold when 1000 unlabeled samples are replayed in CIFAR-100.

**Stability Analysis**   The superiority of our proposed persistence homology distillation lies in its insensitivity to noisy and inaccurate representation. According to the stability of persistence homology [38], we can infer that persistence homology distillation is inherently more robust than feature distillation. We also demonstrate its robustness compared to the similarity distillation in Theorem 1.

**Theorem 1.** *Given the two feature space of point data as $\mathcal{P}_0$ and $\mathcal{P}_1$, for all $p \geq 1$ and $h \in \mathbb{Z}^+$, we have*

$$W_p(Dgm_h(\mathcal{R}(\mathcal{P}_0)), Dgm_h(\mathcal{R}(\mathcal{P}_1))) \leq \binom{M-1}{k}^{\frac{1}{p}} W_p^{pair}(\mathcal{P}_0, \mathcal{P}_1), \qquad (7)$$

*where $Dgm_h(\mathcal{R}(\mathcal{P}_0))$ and $Dgm_h(\mathcal{R}(\mathcal{P}_1))$ are the h-dimensional persistence diagram for the Vietoris-Rips filtration on the point set $\mathcal{P}_0$ and $\mathcal{P}_1$ respectively. $W_p^{pair}$ represents the pair-wise distance between the point set $W_p^{pair}(\mathcal{P}_0, \mathcal{P}_1) = inf_\phi(\sum_{u,v \in \mathcal{P}_0} | \|v - u\| - \|\phi(v) - \phi(u)\| |^p)^{\frac{1}{p}})$, where $\phi$ is a bijection between the two set $\mathcal{P}_0$ and $\mathcal{P}_1$.*

The proof is shown in Appendix A.3. This theorem demonstrates that the persistence homology is robust to the perturbations of similarity between samples and inaccurate sample-wise representation.

# 4   Empirical Study

## 4.1   Experiment Settings

**Datasets.**   We evaluate our method on three datasets with different classes and resolutions. CIFAR-10 [39] is a dataset containing colored images classified into 10 classes, which consists of 5,000 training samples and 1,000 testing samples of size 32 * 32 per class. CIFAR-100 [39] comprises 500 training images and 100 testing images per class, with the same size with CIFAR-10. ImageNet-100 [40], a subset of ImageNet-1000, is composed of 100 classes with 1300 images per class for training and 500 images per class for validation. ImageNet-100 resembles real-world scenes with a higher resolution of 256*256.

**Continual Semi-supervised Setting.**   For both CIFAR-10 and CIFAR-100, we train the models with different levels of supervision, i.e., $\lambda \in \{0.8\%, 5\%, 25\%\}$. For instance, the label ratio corresponds to 4, 25, and 125 annotated samples per class in CIFAR-100. Regarding ImageNet-100, we choose $\{1\%, 7.7\%\}$ for label ratio. To build the continual datasets, we use two learning sequence settings in the literature NNCSL [13] and DSGD [14], and divide the datasets into equally disjoint tasks: 5/10/10 tasks for CIAFR-10/CIFAR-100/ImageNet-100, i.e., 2/10/10 class per task. We also follow the standard incremental setting of CL and assume that only limited space is left for previous data. In other words, the memory buffer can be constructed for both labeled and unlabeled data. Following [13, 7], we set the buffer size as 500 and 2000.

**Evaluation Metrics.**   We evaluate the performance of different methods considering the average accuracy over all the classes seen after the last task $AA_T = \frac{1}{T} \sum_{i=1}^{T} a_{T,i}$, and average incremental accuracy of each task $AIA = \frac{1}{T-1} \sum_{t=2}^{T} AA_t$ [1]. Analysis with backward transfer, which presents the degree of forgetting $BWT = \frac{1}{T-1} \sum_{t=1}^{T-1} (a_{T,t} - a_{t,t})$, is also evaluated in the ablation study. The forgetting is usually reflected as negative $BWT$.

**Baselines.**   We adopt the semi-supervised continual learning pipeline of DSGD [14]. Two categories of methods: distillation-based method iCaRl[4] and parameter consolidation method DER[7] are considered as the strategies for overcoming catastrophic of labeled data. A common semi-supervised learning method Fixmatch [36] is adopted to learn discriminate representation of unlabeled data.

Table 1: Average incremental accuracy and average accuracy of various methods on 5-tasks CIFAR-10, 10-tasks CIFAR-100 and 10-tasks ImageNet-100 settings following the learning sequence [14], with a memory buffer size of 2000. The improvements of PsHD compared to the state-of-the-art methods are highlighted in blue color.

| Method | CIFAR-10 | | | | CIFAR-100 | | | | ImageNet-100 | | | |
|---|---|---|---|---|---|---|---|---|---|---|---|---|
| | 5% | | 25% | | 5% | | 25% | | 1% | | 7.7% | |
| | avg | last | avg | last | avg | last | avg | last | avg | last | avg | last |
| iCaRL[4] | 65.9 | 56.4 | 65.4 | 51.4 | 28.1 | 15.3 | 44.1 | 30.7 | 19.9 | 12.9 | 30.8 | 16.7 |
| iCa_Fix[14] | 83.7 | 79.2 | 82.9 | 78.8 | 49.8 | 31.3 | 56.9 | 41.4 | 26.4 | 15.6 | 37.5 | 21.0 |
| DSGD[14] | 83.8 | 79.1 | 83.2 | 79.0 | 53.4 | 35.9 | 58.1 | 43.1 | 28.3 | 19.1 | 50.5 | 32.1 |
| PsHD | 85.4 | 81.5 | 86.1 | 81.4 | 56.6 | 38.8 | 59.4 | 43.2 | 29.7 | 22.4 | 53.4 | 35.0 |
| Gain | **+1.6** | **+2.4** | **+2.9** | **+2.4** | **+3.2** | **+2.9** | **+1.3** | **+0.1** | **+1.4** | **+3.2** | **+2.9** | **+2.9** |
| DER[7] | 70.4 | 65.6 | 70.9 | 62.3 | 32.8 | 26.5 | 57.2 | 48.8 | 20.8 | 14.4 | 41.2 | 32.6 |
| DER_Fix[14] | 86.1 | 81.2 | 83.9 | 81.4 | 52.0 | 44.5 | 66.7 | 53.6 | 29.8 | 23.3 | 59.5 | 51.0 |
| DSGD[14] | 86.3 | 81.6 | 85.0 | 81.9 | 57.9 | 46.7 | 69.1 | 58.5 | 29.8 | 23.3 | 59.9 | 51.5 |
| PsHD | 86.0 | 81.8 | 88.0 | 83.4 | 58.3 | 47.2 | 69.5 | 58.7 | 30.7 | 24.2 | 60.6 | 52.1 |
| Gain | **-0.3** | **+0.2** | **+3.0** | **+1.5** | **+0.4** | **+0.5** | **+0.4** | **+0.2** | **+0.9** | **+0.9** | **+0.7** | **+0.6** |

Table 2: Average accuracy of different methods test with 5-tasks CIFAR-10 and 10-tasks CIFAR-100 settings following learning sequence of [13] with 500 samples replayed. The data with underline is the best performance within existing methods.

| Method | Source | CIFAR-10 | | | CIFAR-100 | | |
|---|---|---|---|---|---|---|---|
| | | 0.8% | 5% | 25% | 0.8% | 5% | 25% |
| iCaRL[4] | CVPR 2017 | 24.7 | 35.8 | 51.4 | 3.6 | 11.3 | 27.6 |
| FOSTER[6] | ECCV 2022 | 43.3 | 51.9 | 57.1 | 4.7 | 14.1 | 21.7 |
| X-DER[42] | TPAMI 2022 | 33.4 | 48.2 | 58.9 | 8.9 | 18.3 | 23.9 |
| PseudoER[13] | ICCV 2023 | 50.5 | 56.5 | 57.0 | 8.7 | 11.4 | 12.3 |
| CCIC[15] | PRL 2021 | 54.0 | 63.3 | 63.9 | 11.5 | 19.5 | 20.3 |
| PAWS[43] | ICCV 2021 | 51.8 | 64.6 | 65.9 | 16.1 | 21.2 | 19.2 |
| CSL[13] | ICCV 2023 | 64.5 | 69.6 | 70.0 | 23.6 | 26.2 | 29.3 |
| NNCSL[13] | ICCV 2023 | 73.2 | 77.2 | 77.3 | 27.4 | 31.4 | 35.3 |
| DER_Fix[14] | AAAI 2024 | 73.4 | 75.2 | 76.5 | 25.4 | 41.3 | 46.3 |
| DSGD[14] | AAAI 2024 | 73.6 | 76.1 | 77.5 | 25.9 | 41.6 | 47.1 |
| PsHD | Ours | 74.5**+1.1** | 77.6**+0.4** | 78.0**+0.5** | 27.7**+0.3** | 42.4**+0.8** | 47.8**+0.7** |

## 4.2 Quantitative Results

**Comparison with SSCL Methods.** Table 1 shows the disrupted class order following [14, 41] across the three datasets, and also report the average incremental accuracy to demonstrate the effectiveness of the proposed method during the whole evolution. The backbone architectures remain consistent with those used in [14], while the distillation loss on unlabeled samples in the memory buffer is replaced by our newly proposed persistent homology distillation loss. We obtain a general improvement on the two backbones, demonstrating the intrinsic topological features captured by PH are more efficient than the sample-wise local similarity distillation. In addition, the scale of the sub-graph is an important parameter that should be selected in DSGD [14], which does not occur in the proposed PsHD since its automatic multi-scale characteristic. The results of ImageNet-100 also demonstrate the effectiveness on a more challenging benchmark with a higher resolution.

For a more comprehensive comparison, we also validate our method on the default class sequence. Table 2 shows the results of semi-supervised continual learning on CIFAR-10 and CIFAR-100 without shuffle class order. It can be seen that our PsHD outperforms all the methods in the average accuracy across several label ratios on both datasets. As representative relation distillation methods, NNCSL [13] and DSGD [14] show comparative results compared to sample representation distillation. Our method surpasses the two approaches further, indicating the superiority of distilling persistent homology for unlabeled data in semi-supervised continual learning. In addition, NNCSL [13] only

considers the rehearsal of labeled data, limiting its potential advantages on unlabeled data, while our methods focus on the unlabeled data and complement the class-instance similarity preservation.

**Better Adaptability to Unlabeled Data Compared with Distillation Methods.** Table 3 presents a comparison of traditional knowledge distillation and its more recent variants from [48] in the learning tasks of our SSCL problem. The effectiveness of logits and features distillation diminishes generally, particularly as tasks become more challenging and numerous. For instance, the performance on CIFAR100 is lower than that on CIFAR10, and a reduced labeled ratio results in decreased accuracy. Although relation and topology distillation are relatively effective for SSCL, their stability is still inferior to that of our method.

Table 3: Average accuracy of different knowledge distillation methods applied on SSCL.

| Method | Type | CIFAR10 | | CIFAR100 | |
|---|---|---|---|---|---|
| | | 5% | 25% | 5% | 25% |
| iCaRL[4] | logits | 79.2 | 78.8 | 31.3 | 41.4 |
| Foster[6] | logits | 75.0 | 70.4 | 24.6 | 38.8 |
| LUCIR[44] | feature | 75.2 | 74.7 | 32.0 | 32.6 |
| Podnet[45] | feature | 57.9 | 69.0 | 21.4 | 21.1 |
| R-DFIL[46] | relation | 78.5 | 78.8 | 34.7 | 34.2 |
| DSGD[14] | relation | 79.1 | 79.0 | 35.9 | 43.1 |
| TopKD[47] | topology | 78.7 | 79.8 | 35.4 | 41.9 |
| PsHD | topology | **81.5** | **81.4** | **38.8** | **43.2** |

**Stability to Noise Interference.** We also validate the stability of our methods under noise interference. As shown in Table 4, we add Gaussian noise with standard deriviation {0.2, 1, 1.2} on five distillation-based continual learning methods and our PsHD. The results indicate that our method exhibits greater stability in terms of both forgetting degree and accuracy, despite the presence of higher noise levels. These findings further confirm the stability conclusions drawn in Theorem 1.

Table 4: Comparison of distillation methods with Gaussian noise interference on CIFAR10 with 5% supervision. $\sigma$ is the standard deviation.

| $\sigma$ | Podnet[45] | | LUCIR[44] | | R-DFCIL[46] | | DSGD[14] | | TopKD[47] | | PsHD | |
|---|---|---|---|---|---|---|---|---|---|---|---|---|
| | BWT↓ | AA↑ | BWT↓ | AA↑ | BWT↓ | AA↑ | BWT↓ | AA↑ | BWT↓ | AA↑ | BWT↓ | AA↑ |
| 0.2 | 31.8 | 55.6 | 21.8 | 71.5 | 19.2 | 77.5 | 18.1 | 76.7 | 19.7 | 76.5 | **14.0** | **78.7** |
| 1.0 | 44.4 | 27.9 | 34.3 | 57.3 | 28.2 | 62.1 | 27.1 | 64.2 | 23.0 | 67.2 | **17.7** | **70.9** |
| 1.2 | 49.8 | 34.6 | 36.4 | 56.4 | 26.4 | 61.8 | 23.7 | 64.1 | 22.8 | 65.8 | **18.3** | **67.8** |

## 4.3 Parameter Analysis

**Ablation Study.** We firstly ablate the components of persistence homology distillation through quantitative analysis in Table 5 on both CIFAR-10 and CIFAR-100 across 5% and 25% label ratios. The baseline is based on iCaRL_Fix and DER_Fix, where the iCaRL and DER are two methods in traditional continual learning and Fix is Fixmatch for semi-supervised learning. $\mathcal{L}_{SSL}$ and $\mathcal{L}_{hd}$ represent the semi-supervised loss and homology diagrams distillation loss, respectively. The general improvements of our structural distillation loss confirm the contributions of our proposed components.

**Visualization of the Effectiveness of Persistence Homology Distillation.** We also provide the class activation heatmap of old categories in the initial and following tasks during the continual learning process in ImageNet-100. These heatmap pictures localize class-specific discriminative regions, as shown in Figure 4. It can be seen from (a) part of the three groups that the model without our persistence homology distillation usually drop the class-specific attention region with the task takes advancement, despite the init model having learned a discriminative representation. In addition, the base model may even shift focus on the background area in the middle tasks, such as Task 6-9 in 2.a, resulting in unreliable knowledge accumulation. This is because of the insufficient previous data support, memory data under-utilization, and feature space transformation, which lead to attention shift as shown in 1.a, and even attention vanishment as shown in 3.a.

In contrast, the introduction of our persistence homology distillation is able to preserve the discriminative region along with the whole continual learning, as depicted in (b) part, which indicates that preserving persistence homology that captures the class-wise structural information encourages the model to focus on the class-specific region continually. It is worth noting that even if the initial model is not capable of noticing the class-specific region, as shown in 3.a, which may result from insufficient supervision, our method still empowers the following model training to activate the potential class-relevant region since it focuses on the global distribution of each class and pushes the

Table 5: Ablation study of proposed persistence homology distillation

| Method | $\mathcal{L}_{SSL}$ | $\mathcal{L}_{hd}$ | CIFAR10_5% | | CIFAR10_25% | | CIFAR100_5% | | CIFAR100_25% | |
|---|---|---|---|---|---|---|---|---|---|---|
| | | | avg | last | avg | last | avg | last | avg | last |
| iCaRL | $\checkmark$ | | 83.7 | 79.2 | 82.9 | 78.8 | 54.2 | 36.1 | 58.4 | 41.1 |
| | $\checkmark$ | $\checkmark$ | $85.3_{+1.5}$ | $80.8_{+1.6}$ | $85.4_{+2.6}$ | $80.5_{+1.7}$ | $56.4_{+2.2}$ | $38.8_{+2.7}$ | $59.2_{+0.8}$ | $42.2_{+1.1}$ |
| DER | $\checkmark$ | | 86.1 | 81.2 | 84.9 | 81.4 | 57.8 | 46.3 | 68.1 | 57.2 |
| | $\checkmark$ | $\checkmark$ | $86.2_{+0.1}$ | $81.8_{+0.6}$ | $87.9_{+3.0}$ | $83.4_{+2.0}$ | $58.2_{+0.4}$ | $47.2_{+0.9}$ | $68.3_{+0.2}$ | $57.7_{+0.5}$ |

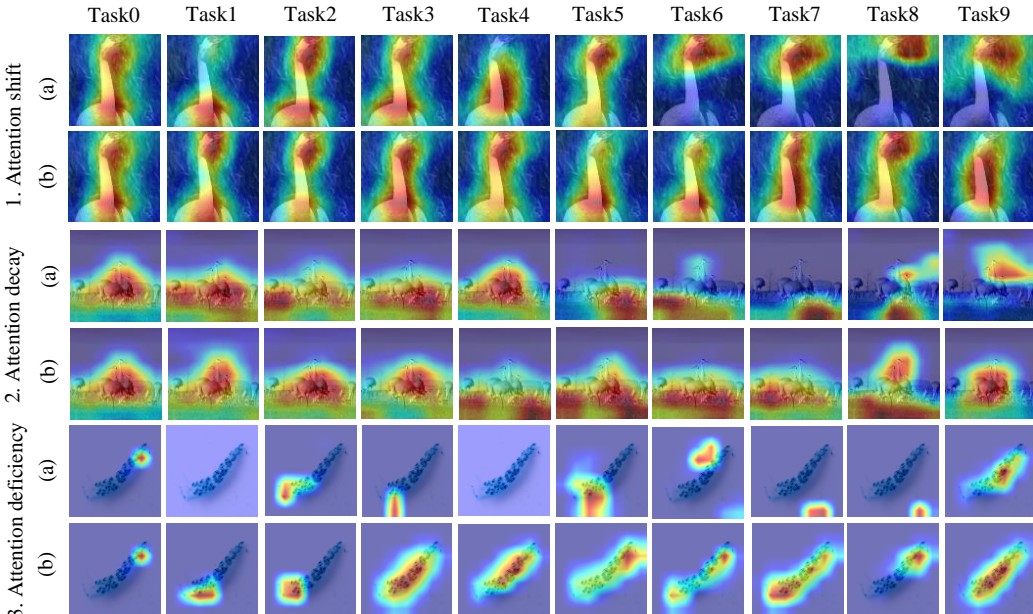

Figure 4: Visualization of activation heatmap during the continual learning process, where the categories belong to Task0. 1-3(a) correspondence to PsHD without the $L_{hd}$, and 1-3(b) correspondence to PsHD with $L_{hd}$. The red area localizes class-specific discriminative regions.

semantic similar samples to be related in representation, as shown in 3.b. Additional visualization of PsHD is depicted in Appendix A.5.

**Effect of the Weight of Persistence Homology Loss.** We evaluate the influence of loss weight $\lambda$ in Eq. 3 across the three datasets, and report the results in Figure 5. It can be seen that a higher weight indicates higher average incremental accuracy and a lower forgetting degree generally. In addition, our method improve the baseline generally within a reasonable range. Too much higher weight on persistence homology loss, for instance bigger than 1, is not supposed to be better for the trade-off problem between old and new tasks. We choose 1 for CIFAR10 and CIFAR100, and 1.5 for ImageNet100 in our experiments.

**Effect of k-simplex in Persistence Homology.** The persistence diagram calculation considers the lift-span of $h$-dimensional holes that existed in the nested simplicial complexes data distribution. We verify the effectiveness of dimension $h$ in the homology diagram by considering 0-dimensional holes and both 0,1-dimensional holes in CIFAR-10 and CIFAR-100. The results are shown in Figure 5. It can be seen that considering the persistence of both 0,1-dimensional holes are more effective in CIFAR-10, while CIFAR-100 slightly prefers the simple 0-dimensional holes persistence preserved.

**Effect of Memory Buffer Distribution.** To investigate the necessity of replaying the unlabeled data, we evaluate the effect of memory buffer allocation of labeled and unlabeled data on CIFAR-100 of 5% and 25% label ratios in Table 6, where ratios represent the percentage of labeled data in the memory buffer. The general improvements indicate the percentage of labeled data has an important effect on the performance. It can be observed that replacing 20%-40% labeled samples with unlabeled samples improves 1.49%, 1.27% in 5%. We experimentally choose 0.6-0.9 in the following settings.

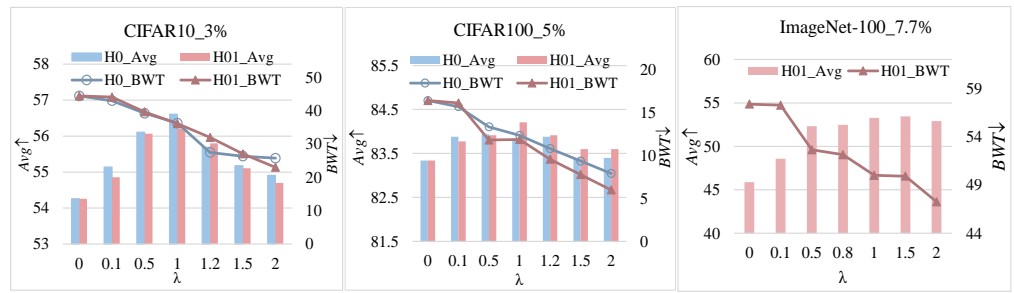

Figure 5: Effectiveness of h-simplex features in persistent homology. H0_Avg and H01_Avg represent the average incremental accuracy based on considering 0-dimensional holes and 0,1-dimensional holes persistence. BWT evaluates the forgetting degree.

Table 6: Effect of data allocation in memory buffer

|  | 1 | 0.9 | 0.8 | 0.7 | 0.6 | 0.5 | 0.4 | 0.3 | 0.2 | 0.1 |
|---|---|---|---|---|---|---|---|---|---|---|
| 5% | 53.94 | 54.60 | **55.43** | 54.63 | 55.21 | 54.45 | 54.57 | 54.30 | 54.21 | 54.52 |
| 25% | 65.32 | **65.82** | 65.48 | 65.45 | 65.50 | 65.44 | 65.17 | 65.3 | 65.00 | 56.10 |

## 4.4 Efficiency Analysis

**Memory Reduction.** Our method is also memory-efficient for SSCL. As shown in Table 7, the proposed PsHD that incorporates unlabeled data surpasses methods relying solely on labeled data, such as NNCSL, by an average of 3.9% across six SSCL settings. To further verify the memory efficiency, we reduced the memory buffer size by 60% to 2000, achieving an average improvement of 1.5% compared to NNCSL across the six semi-supervised continual learning tasks.

Table 7: Average accuracy of different methods following learning sequence of [13] with memory buffer size 5120. * represents that the size of memory buffer is 2000.

| Method | CIFAR-10 | | | CIFAR-100 | | |
|---|---|---|---|---|---|---|
|  | 0.8% | 5% | 25% | 0.8% | 5% | 25% |
| PseudoER[13] | 55.4 | 70.0 | 71.5 | 15.1 | 24.9 | 30.1 |
| CCIC[15] | 55.2 | 74.3 | 84.7 | 12.0 | 29.5 | 44.3 |
| CSL[13] | 64.3 | 73.1 | 73.9 | 23.7 | 41.8 | 50.3 |
| NNCSL[13] | 73.7 | 79.3 | 81.0 | 27.5 | 46.0 | 56.4 |
| PsHD | **82.3** | **82.5** | **87.2** | **28.9** | **47.7** | **57.5** |
| PsHD * | 74.6 | 81.5 | 84.6 | 28.3 | 46.0 | 56.5 |

**Computation Efficiency.** A newly topology-based distillation method TopKD [47], which is similar with our method, is proposed for model compression. We prove that the complexity proportion of TopKD and our method is $b^2$, where b is the number of old classes. The specific derivation is clarified in Appendix A.4. This value indicates the relative complexity of TopKD grows polynomially when the class number increases, highlighting our computation superiority.

We evaluate the memory storage, model parameters and training time between our methods and state-of-the-art methods in Appendix A.4 Table 8, indicating the superiority. Despite the acceleration algorithm, a limitation of our method is that the computation of the persistence diagram relies on the Guidh package, which operates on the CPU. We plan to explore the learnable network RipsNet [49] to approximate the persistence diagram and further accelerate the computation in the following work.

## 5 Conclusion

We proposed Persistence Homology Distillation (PsHD) to address catastrophic forgetting and noise sensitivity in Semi-Supervised Continual Learning (SSCL). First, we utilized persistence homology to capture stable topological feature representation of unlabeled data in semi-supervised learning. Next, we proposed a novel persistence homology distillation strategy for SSCL that is insensitive to noise information interference based on the intrinsic topological features and devised an accelerating algorithm to reduce computation costs. Finally, general improvements in three benchmarks demonstrated the efficiency of PsHD in overcoming catastrophic forgetting in SSCL.

## Acknowledgments and Disclosure of Funding

This work was supported in part by the National Science and Technology Major Project under Grant 2022ZD0116500, in part by the National Natural Science Foundation of China under Grants 62436002, 62476195, 61925602, U23B2049, 62222608, and 62406219, in part by Tianjin Natural Science Funds for Distinguished Young Scholar under Grant 23JCJQJC00270, in part by Zhejiang Provincial Natural Science Foundation of China under Grant LD24F020004, and in part by China Postdoctoral Science Foundation - Tianjin Joint Support Program under Grant Number 2023T014TJ.

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

# A    Appendix / supplemental material

**Content**

## A.1    Preliminaries

**Persistence Homology.**    Now we explain how persistence homology works. It is easy to obtain an inclusion map $K_i \hookrightarrow K_j, 0 < i < j < m$ from Eq. 2. These maps also induce a group homomorphism of the corresponding homology groups

$$F_d^{i,j} : H_d(K_i) \mapsto H_d(K_j), 0 < i < j < m, \tag{8}$$

where $H_d(K_i)$ and $H_d(K_j)$ are homology groups of the $d^{th}$ order. A homology class $h$ is born at time $f_i$ if $h \in H_d(K_i)$ but $h \notin \text{img}(F_d^{i-1,j})$. Provided $h$ is born at $f_i$, $h$ dies at time $f_j$ only if $F_d^{i,j-1}(h) \notin \text{img}(F_d^{i-1,j-1})$ while $F_d^{i,j}(h) \in \text{img}(F_d^{i-1,j})$ [50]. ('img' refers to the image of a map). These maps reveal which feature features persist (a feature refers to a homology class, corresponding to a component or a n-dimensional hole). An easier way to understand is as follows. A new homology class is born at time $f_i$ only if it does not appear at the existing features, and it dies at time $f_j$ only if it still persists at time $f_{j-1}$ and is merged into other features at time $f_j$. The time between death and birth of a feature is called its lifetime or persistence.

## A.2    Algorithm of Persistence Homology Distillation

We illustrate our persistence homology distillation in Algorithm 1.

---
**Algorithm 1** Persistence Homology Distillation
---
**Input**: Replayed batch of unlabeled data: $B_u^o$.
**Parameter**: Embedding encoder of current task $f_{new}$ and old task $f_{old}$,
1:  $E \leftarrow B_u^o, i \leftarrow 0, S \leftarrow \emptyset$
2:  **repeat**
3:      $x_i \leftarrow \text{RandomSample}(E)$
4:      $(f_{x_i}^{new}, f_{x_i}^{old})$
5:      Group the weighted k-hop neighborhood set $\mathcal{N}(x_i, k)$ by $f_{x_i}^{old}$
6:      $E \leftarrow E \backslash \mathcal{N}(x_i, k), S \leftarrow S \cup x_i, i \leftarrow i + 1$
7:      Compute the Wasserstein distance $d_{\mathcal{N}_x^k}(f_{old}, f_{new})$ according to Eq. 5
8:  **until** $E$ is empty
9:  Minimize the total loss $\mathcal{L}_{hd} = \frac{1}{|S|} \sum_{x \in S} d_{\mathcal{N}_x^k}(f_{old}, f_{new})$

---

## A.3    Proof of Stability Theorem

**Theorem 1.**    Given the two feature space of point data as $\mathcal{P}_0$ and $\mathcal{P}_1$, for all $p \geq 1$ and $h \in \mathbb{Z}^+$, we have

$$W_p(Dgm_h(\mathcal{R}(\mathcal{P}_0)), Dgm_h(\mathcal{R}(\mathcal{P}_1))) \leq \binom{M-1}{k}^{\frac{1}{p}} W_p^{pair}(\mathcal{P}_0, \mathcal{P}_1), \tag{9}$$

where $Dgm_h(\mathcal{R}(\mathcal{P}_0))$ and $Dgm_h(\mathcal{R}(\mathcal{P}_1))$ are the h-dimensional persistence diagram for the Vietoris-Rips filtration on the point set $\mathcal{P}_0$ and $\mathcal{P}_1$ respectively. $W_p^{pair}$ represents the pair-wise distance between the point set $W_p^{pair}(\mathcal{P}_0, \mathcal{P}_1) = \inf_\phi(\sum_{u,v \in \mathcal{P}_0} | \, \|v - u\| - \|\phi(v) - \phi(u)\| \, |^p)^{\frac{1}{p}})$, where $\psi$ is a bijection within the same set, such as $\mathcal{P}_0 \to \mathcal{P}_0$ or $\mathcal{P}_1 \to \mathcal{P}_1$, and $\phi$ is a bijection between the two set $\mathcal{P}_0$ and $\mathcal{P}_1$.

*Proof.* Let $\phi : \mathcal{P}_0 \to \mathcal{P}_1$ be the bijection which achieves the minimum of

$$W_p^{pair}(\mathcal{P}_0, \mathcal{P}_1) = \inf_{\phi} \left( \sum_{u,v \in \mathcal{P}_0} \left| \|u - v\| - \|\phi(u) - \phi(v)\| \right|^p \right)^{\frac{1}{p}}$$

Relabel the points in $\mathcal{P}_0 = \{x_1, \ldots, x_M\}$ and $\mathcal{P}_1 = \{y_1, \ldots, y_M\}$ so that $\phi(x_i) = y_1$. Let K be the complete simplicial complex on $M$ vertices $\{v_1, \cdots, v_M\}$. Define functions $f, g : K \to \mathbb{R}$ by $f([v_{i_0}, v_{i_1}, \cdots, v_{i_k}])$ the time when $[x_{i_0}, x_{i_1}, \cdots, x_{i_k}]$ is included in $\mathcal{R}(\mathcal{P}_0)$ and $g([v_{i_0}, v_{i_1}, \cdots, v_{i_k}])$ the time when $[y_{i_0}, y_{i_1}, \cdots, y_{i_k}]$ is included in $\mathcal{R}(\mathcal{P}_1)$.

Suppose $h \geq 1$, then

$$
\begin{aligned}
|f([v_{i_0}, v_{i_1}, \cdots, v_{i_k}]) - g([v_{i_0}, v_{i_1}, \cdots, v_{i_k}])| &= \left| (\max_{j,l}\{\|x_{i_j} - x_{i_l}\|\} - \max_{j,l}\{\|y_{i_j} - y_{i_l}\|\} \right| \\
&\leq \max_{j,l} \left| \|x_{i_j} - x_{i_l}\| - \|y_{i_j} - y_{i_l}\| \right| \\
&\leq \max_j \sum_l \left| \|x_{i_j} - x_{i_l}\| - \|y_{i_j} - y_{i_l}\| \right|.
\end{aligned}
$$

Since $K$ is the complete simplicial complex over $M$ vertices, each edge $[v_i, v_j]$ appears in $\binom{M-2}{k-1}$ $h$-simplices, we only need to decide which extra $k$-1 vertives to include.

Based on the cellular stability theorem [38],

$$
\begin{aligned}
W_p&(Dgm_h(\mathcal{R}(\mathcal{P}_0)), Dgm_h(\mathcal{R}(\mathcal{P}_1)))^p \\
&\leq \sum_{[v_{i_0}, \cdots, v_{i_k}]} |f([v_{i_0}, v_{i_1}, \cdots, v_{i_k}]) - g([v_{i_0}, v_{i_1}, \cdots, v_{i_k}])|^p \\
&\quad + \sum_{[v_{i_0}, \cdots, v_{i_{k+1}}]} |f([v_{i_0}, v_{i_1}, \cdots, v_{i_{k+1}}]) - g([v_{i_0}, v_{i_1}, \cdots, v_{i_{k+1}}])|^p \\
&\leq \sum_{[v_{i_0}, \ldots, v_{i_k}]} \max_{j,l} \left| \|x_{i_j} - x_{i_l}\| - \|y_{i_j} - y_{i_l}\| \right|^p \\
&\quad + \sum_{[v_{i_0}, \cdots, v_{i_{k+1}}]} \max_{j,l} \left| \|x_{i_j} - x_{i_l}\| - \|y_{i_j} - y_{i_l}\| \right|^p \\
&\leq \sum_{[v_{i_0}, \ldots, v_{i_k}]} \max_j \sum_{l \in [v_{i_0}, \cdots, v_{i_k}]} \left| \|x_{i_j} - x_{i_l}\| - \|y_{i_j} - y_{i_l}\| \right|^p \\
&\quad + \sum_{[v_{i_0}, \cdots, v_{i_{k+1}}]} \max_j \sum_{l \in [v_{i_0}, \cdots, v_{i_{k+1}}]} \left| \|x_{i_j} - x_{i_l}\| - \|y_{i_j} - y_{i_l}\| \right|^p \\
&\leq \sum_{x_j \in \mathcal{P}_0} \binom{M-2}{k-1} \sum_{x_l \in \mathcal{P}_0} \left| \|x_j - x_l\| - \|y_j - y_l\| \right|^p \\
&\quad + \sum_{x_j \in \mathcal{P}_0} \binom{M-2}{k} \sum_{x_l \in \mathcal{P}_0} \left| \|x_j - x_l\| - \|y_j - y_l\| \right|^p \\
&\leq \binom{M-1}{k} W_p^{pair}(\mathcal{P}_0, \mathcal{P}_1)^p.
\end{aligned}
$$

For $h = 0$ the calculations are easier as the vertex values are all 0.

$$
\begin{aligned}
W_p(Dgm_0(\mathcal{R}(\mathcal{P}_0)), Dgm_0(\mathcal{R}(\mathcal{P}_1))) &\leq \sum_{i<j} |f([v_i, v_j]) - g(v_i, v_j)|^p \\
&= \sum_{i<j} \left| \|x_i - x_j\| - \|y_i - y_j\| \right|^p \\
&\leq W_p^{pair}(\mathcal{P}_0, \mathcal{P}_1)^p.
\end{aligned}
$$

## A.4 Efficiency of Proposed Algorithm

As the similar topology-based distillation methods, ours computation cost is smaller than the TopKD [47]. The main reason is because of the smaller number of persistence homology points $u$ in the persistence homology $Dgm_h(\mathcal{N}_x^k, f_{old}) \cup \triangle$ of Eq. 5. The number of $u$ is proportional to simplices number $n$ since HP complexity is $\mathcal{O}(n^{2.5})$. It is hard to approximate $n$, while we can deriviate its upper bound in our method $\mathcal{U}_{n,PsHD}^K = b \sum_{i=1}^{K} \binom{M/b}{i}$ and TopKD $\mathcal{U}_{n,TopKD}^K = \sum_{i=1}^{K} \binom{M}{i}$, where $M$ is the replayed samples in each batch, $b$ is approximate to class number and $K$ is the highest number of distilled simplex. We experimentally set $K$ to be 3, since 1-simplex derive when $i = 2$ and dead when $i = 3$. The upper bound ratio is $b^2$.

*Proof.*

$$\mathcal{U}_{n,TopKD}^3 = M + \frac{M(M-1)}{2} + \frac{M(M-1)(M-2)}{6},$$

$$\mathcal{U}_{n,PsHD}^3 = M + \frac{M(M/b-1)}{2} + \frac{M(M/b-1)(M/b-2)}{6},$$

$$\frac{\mathcal{U}_{n,TopKD}^3}{\mathcal{U}_{n,PsHD}^3} = \frac{6b^2 + (M-1)3b^2 + (M-1)(M-2)b^2}{6b^2 + (M-b)3b + (M-b)(M-2b)}$$
$$= \frac{5b^2 + M^2b^2}{5b^2 + M^2}$$
$$\approx b^2 (b \to M, M >> 5).$$

When the number of old class $b$ raises in the continual learning process, the relative computational complexity grows quadratically with $b$. Heavy computation usually accumulates during the whole learning process, so this ratio demonstrating the superiority of our methods.

In addition, we also evaluate the memory storage, model parameters and training time between our methods and state-of-the-art methods in Table 8. When reducing the replayed examplers to 2000, our method still achieves the best performance 46.0%. Compared to the best indices, our method lessen the training time by 10.9%, example buffer by 60.9%, and memory size by 28.4%. Therefore, while existing strong baselines excel in different aspects, NNCSL in accuracy, DER_Fix in training time, and DSGD relatively balanced computation costs and accuracy, our method achieves the overall least resources overhead and the highest accuracy.

Table 8: Effectiveness and efficiency comparison on CIFAR100_5%. The memory size is the sum of the storage of model parameters and replayed examples.

| | Train.(h) | Parameters(m) | Example | Memory Size (MB) | Acc(%) |
|---|---|---|---|---|---|
| NNCSL | 12.1 | 11.8 | 5120 | 56.58 | 46.0 |
| DER_Fix | 9.1 | 4.6 | 5120 | 30.80 | 43.61 |
| DSGD | 9.8 | 4.6 | 5120 | 30.80 | 44.61 |
| PsHD | 14.2 | 4.6 | 5120 | 30.80 | **47.7** |
| PsHD | **8.1** | 4.6 | **2000** | **22.07** | 46.0 |

## A.5 Visualization of Effectiveness of PsHD

We provide additional visualizations of PsHD in Figure 6. Expect for the issue in Figure 4, the attention bias shows that the initial model fails to recognize the accurate class-specific regions, making it difficult for subsequent models to consistently focus on the correct areas. By incorporating persistent homology features, the model is encouraged to direct attention to the appropriate regions.

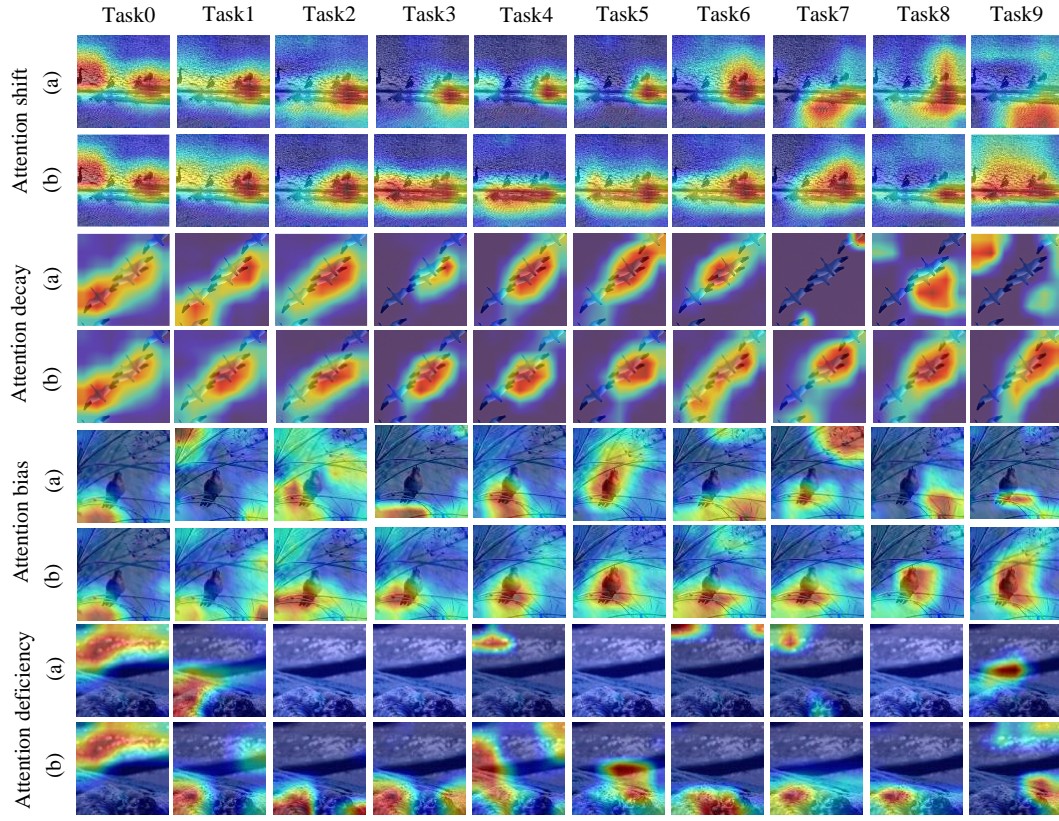

Figure 6: Visualization of activation heatmap during the continual learning process, where the categories belong to Task0. (a) part correspondence to PsHD without the $L_{hd}$, and (b) part correspondence to PsHD with $L_{hd}$. The red area localizes class-specific discriminative regions.

