# OpenReview forum: "Persistence Homology Distillation for Semi-supervised Continual Learning"
_NeurIPS.cc/2024/Conference — NeurIPS 2024 poster_

### Official Review · Reviewer_2waA · 2024-07-09

**Soundness:** 3
**Presentation:** 1
**Contribution:** 2
**Rating:** 5
**Confidence:** 4

**Summary:**

The paper proposes a new method PsHD to preserve intrinsic structural information in semi-supervised continual learning. The method proposes to uses distillation and cross-entropy loss on the continual learning samples.

**Strengths:**

1. I think the paper presents quite comprehensive experiments with different settings. In all of the experiments, the work demonstrate at least marginal improvement compared to previous methods
2. I think figure.3 looks very interesting and seems to justify the paper's motivation.

**Weaknesses:**

1. I think the paper can be further polished in terms of writing, presentation and clarity. For example. I think Figure.4 can be further improved for clarity.
2. The results raise some concerns regarding the proposed method. In Table 4, we observe that changing the data allocation ratio results in performance change within 1.5% for 5% and within 1% for 25% setting. Also in Table.4, optimal data ratio are different for the two settings. We also observe in Table 1 that the proposed method leads the previous methods by less than 1%. It makes the author's claim for their ``superiority'' performance quite unsupported.
3. This brings my third concern to the work: Some language in the paper seems overclaimed. The paper tried to adopt many large words like ``significant'' and ``superior'' while the performance does not support such claims.
4. In Figure.4, larger lambda seem to help the model performance, then why not train with larger lambda? Maybe there is an elbow effect but there needs more experiment to show that.

**Questions:**

Please in weakness

---

> ### Author Rebuttal · Authors · 2024-08-07
>
> **A1.** We will make substantial revisions to enhance the overall clarity and readability:
>
> **(1) Provide detailed results to verify the description of our advantages.** As shown in Table 1. of attached PDF file, we provide additional comparison of the forgetting rate  (BWT) among strong baselines and our method to demonstrate its effectiveness in alliviate catastrophic forgetting. Our method achieves 3.4% forgetting mitigation with 0.9% accuracy increasement on average  compared with DSGD, and  18\% forgetting mitigation with 6.15\% accuracy increasement on average compared to NNCSL. Moreover, the effeciency comparison is also provided in Table 2, and the stability contrast to noise interference is supplemented in Table 4.
>
> **(2) Redesigned Figure 4 to make it more informative and readable.** As shown in Figure 1 of the attached PDF file, we conducted additional ablation experiments on larger distillation loss weight $\lambda$, specifically  {1.2, 1.5, 2} to further verify the effect of proposed homology distillation loss in overcoming catastrophic forgetting. We selecte the optimal  weights $\lambda=1$ for the three dataset that  balance the forgetting (BWT) and average incremental accuracy (Avg) best.  Additionally, we expand the ablation study of persistence homology dimension to determine appropriate homology featuresfor distillation, using 0-simplex for the simple dataset CIFAR10, and 0,1-simplex for complex data, such as CIFAR100 and ImageNet-100. Lastly, we added legends to each diagram for better readability.
>
>
> **A2.**
> Selecting an optimal ratio of labeled to unlabeled data in a limited memory buffer to represent an entire dataset is an open problem in semi-supervised continual learning with different degrees of supervision. We firstly offer experimentally valid allocation ratios for several benchmarks.
> Regarding the limited improvements of Table 1, we enhance the results by providing the degree of forgetting (BWT). We further provide evidence of the efficiency and stabilization of PsHD compared to newly strong baselines.
>
>
> **(1) First consideration of ideal allocation labeled vs. unlabeled ratio in SSCL.** Our experiments indicate that indiscriminate utilization of unlabeled data can diminish accuracy.  As shown in Table 2 of the PDF file, the introduction of logits distillation on unlabeled data (iCarL_U) leads to 4.97% decrease on average compared to distillation only on labeled data (iCaRL). These diminishes are different across supervision ratios, since higher supervision can ensure confident representation of unlabeled data. Therefore, the ideal allocation ratio remains unresolved in existing SSCL methods. We provide experimentally valid allocation ratios for labeled data allocation ratios between 0.6 and 0.9 for a general reproducibility and transferability.
>
> **(2) Enhanced forgetting rate in Table 1 and better knowledge distillation on unlabeled data.**  As shown in Table 1 of the attached PDF file, our method surpasses the NNCSL by 6.15\% with 18\% forgetting on average, outperforms DER\_Fix by 1.63\% with 6.52\% forgetting on average, and exceeds DSGD by 0.89\% with 3.35\% forgetting on average.
> Moreover, our methods exhibit better adaption in distillation knowledge in unlabeled data, achieving a 35.2\% reduction in training time, a 60.9\% reduction in examples buffer, and a 28.4\% memory size reduction compared to the best indexes in CIFAR100\_5\%.
>
> **(3) More stable distillation for SSCL.** The training stability of our method can be verified through comprehensive ablation experiments. As shown in Table 4 of the attached PDF file, we add Gaussian noise with weight \{0.2,1,1.2\} to five typical distillation-based continual learning methods: iCaRL, Foster, Podnet, LUCIR, and R-DFCIL. Our method demonstrates stability in both the degree of forgetting and accuracy, even under higher noise levels. These results are consistent with our Theorem 1.
>
> To sum up, our method PsHD effectively mitigates catastrophic forgetting of unlabeled data and achieves a better trade-off between adapting old tasks and preserving new tasks in SSCL.
>
> **A3.** We include more detailed experimental results to emphasize the strengths of our method and validate our conclusions. Additionally, to ensure rigor, we will also improve the modifier of experiment results and illustrate the limitations at the same time. Such as use 'comparable' or 'competitive' to illustrate  the accuracy improvement less than 1%, and clarify the limited improvement on 500 replayed samples.
>
>
> **A4.** We supplement more experiments to validate the effect of distillation weight lambda value and persistence dimension H0 and present the results in Figure 1 of the attached PDF file. Higher distillation weight results in lower forgetting (BWT) and higher average incremental accuracy (Avg). The optimal persistence homology distillation weights are also highlighted in each benchmark. Additionally, we improve the arrangement of Figure 4 for better readability.

---

> > ### Comment · Reviewer_2waA · 2024-08-14
> >
> > I thank the authors for their rebuttal. After considering the rebuttal and other reviewers' comments, I will raise my score as the rebuttal addresses my concerns.

---

> > > ### Author Response · Authors · 2024-08-14
> > >
> > > We sincerely appreciate the reviewer's response. Meanwhile, we are very grateful for the reviewer's recognition of our responses and work, while increasing the final rating to ‘borderline accept’. Great thanks again!

---

### Official Review · Reviewer_K7Tp · 2024-07-10

**Soundness:** 2
**Presentation:** 3
**Contribution:** 2
**Rating:** 5
**Confidence:** 3

**Summary:**

The paper proposes a persistence homology knowledge distillation for continual learning (PsHD). PsHD loss is calculated using a ''memory buffer'' between a previous variant of a network and a new one. Experiments show some improvement w.r.t. baselines. Ablation studies are provided.
The main issue of the paper is limited novelty, also some essential details are missing (see below).

**Strengths:**

1. Novelty: this is a first application of persistent homology to continual learning.
2. Experiments are correct, ablation studies are provided.
3. The manuscript is well organized, the idea is easily comprehensible.
4. Visualizing of attentions maps helps to reveal how the method improves continual learning.

**Weaknesses:**

1. The idea that knowledge distillation can help continual learning is not new, see [2]. Given this, the novelty of the paper is small.
Do you have an ablation with a traditional KD [1] and its more recent variants from [2]?
2. A relevant reference [3] is missing. How you paper is related to [3], is your method better?
3. I can't find an explicit equation for $L_{CL}$.

[1] Hinton, G., Vinyals, O., & Dean, J. (2015). Distilling the knowledge in a neural network. arXiv preprint arXiv:1503.02531.
[2] Li, S., Su, T., Zhang, X., & Wang, Z. (2024). Continual Learning with Knowledge Distillation: A Survey. Authorea Preprints.
[3] Kim, J., You, J., Lee, D., Kim, H. Y., & Jung, J. H. Do Topological Characteristics Help in Knowledge Distillation?. In Forty-first International Conference on Machine Learning.

**Questions:**

1. How do you differentiate PsHD?
2. You claim that PsHD helps to handle noise, but there is no noise in experimental settings.
The claim "however, traditional distillation strategies often fail in unlabeled data with inaccurate or noisy information" is not proved also.
3. What is the difference between a memory buffer and unlabeled data? To my understanding, for PsHD loss one only needs unlabeled data.
4. Is the SSL part of loss a critical part of your method? Is seems to be completely independent of the main idea (PsHD loss) and, probably, gives to your method an artificial advantage w.r.t. others.

**Other**

1. In Figure 2, two diagrams seem to be identical, while the top simplicial complex has one hole, but the bottom one - two holes.
2. The Wasserstein-p distance (line 164) equation is not correct, because the subscript $\infty$ relates to Wasserstein-$\infty$ distance.
3. line 214: typo CIAFR

**Limitations:**

-

---

> ### Author Rebuttal · Authors · 2024-08-07
>
> **A1.** Traditional CL methods are not always effective for unlabeled data, as they assume the knowledge of previous models is accurate. This specific challenge of SSCL has been explored in previous SSCL methods (NNCSL, DSGD, etc). As suggested by the reviewer, our ablation experiments provide additional verification of this motivation for overcoming catastrophic forgetting of unlabeled data.
>
> **(1)	Negative influence with Traditional KD [1] on SSCL.**  The performance decreases when employing traditional KD [1] directly on SSCL, which is the iCaRL method in our comparison method. As shown in Table 2 of PDF file, the introduction of logits distillation on unlabeled data (iCarL\_U) leads to 4.97\% decrease on average compared to distillation only on labeled data (iCaRL). We also conduct pseudo-label distillation on unlabeled data based on DER, resulting in 5.23\% accuracy decrease (DER\_U) on average. This indicates that inaccurate representation preservation has a negative effect on SSCL.
>
> **(2) Diminished effectiveness of KD variants from [2] for SSCL.** Table 3 of PDF file provides comprehensive ablation experiments of traditional KD [1] and its more recent variants from [2] on our SSCL problem. The effectiveness of logits and feature distillation diminishes, especially as tasks become harder and more numerous, such as in CIFAR100\_5\%.
> Although relation and topology distillation are relatively effective for SSCL, their stability, computation costs, and accuracy require better balancing.
>
> Given the limitations of existing KD methods in SSCL, an adaptive knowledge distillation strategy for SSCL should be further proposed.
>
> **A2.** We acknowledge we didn't seen the reference [3] TopKD, so the citation and comapration are missed even our method is better. The idea of topology feature distillation is similar while ours requires less computation and is more adaptive to SSCL.
>
> **(1) Lower Computation Costs.** Heavy computation usually accumulates during the whole learning process.  Our lower superior computation cost is mainly because of  smaller number of persistence homology points in the PH, which is $u$ in Equation (5). The number of $u$ is proportional to simplices number $n$ since HP complexity is $\mathcal{O}(n^{2.5})$. $n$ is hard to approximate, while we can compute the its upper bound in our PsHD $\sum_{t=0}^{M/b}\binom{M/b}{t}$ and TopKD $\sum_{t=0}^{M}\binom{M}{t}$, where $M$ is the replayed samples in each batch and $b$ is approximate to class number. The ratio is $2^{M-M/B}$, indicating a significant reduction of distillation computation.
>
> **(2) Better Adaptability to Unlabeled Data.**  As shown in Table 3 of PDF file, applying TopKD to CIFAR10\_5\% based on iCaRL results in a decrease in accuracy from 79.2\% to 78.7\%, indicating TopKD's reliance on accurate representations. In contrast, our method improves iCaRL and outperforms TopKD by an average of 2.28\% across four benchmarks. Furthermore, our training time on CIFAR10\_5\% is 7.1 hours, compared to TopKD's 16.5 hours, reducing training time by 56.9\%.
>
> **A3.** $L_{Cl}$ is the continual learning loss, which corresponds to the persistence homology distillation loss in our method. It varies among different types of CL methods. We will change $ L\_{Cl}$  to $L\_{hd}$  to avoid confusion.
>
> **A4.** According to the loss of Equation (5), PsHD is back propagated along with the persistence point $u$ in the persistence diagram. So the gradient backpropagation flow is:
> $L_{hd}\rightarrow u=[u_s,u_e]\rightarrow[dia(\sigma_s), dia(\sigma_e)]\rightarrow [dia(v_1,v_2), dia(v_1,v_2,v_3)]$.
>
> **(1) Illustration of differentiating PsHD.** $u=[u_s,u_e]$ represents a lifespan of one simplex $\sigma_s=\\{v_1,v_2\\}$ exists in $u_s=dia(\sigma_s)$ and disappear or fused to higher dimension simplex $\sigma_e=\\{v_1,v_2,v_3\\}$ in $u_e=dia(\sigma_e)$. $dia(\sigma)$ is the diameter of simplex $\sigma$, which is the maximum distance of any two samples in $\sigma$, and $v_i$ represents the sample. The loss is finally optimized to update the representation.
>
> **(2) End-to-End training of our PsHD.** The origin Gudhi hasn’t been developed on pytorch, so the end-to-end training is unprocurable directly. Our method achieves this progress by making the differentiation on PsHD along the chosen samples.
>
> **A5.** We supplement further ablation experiments on noise interference. As shown in Table 4 of attached PDF file, we add Gaussian noise with standard deriviation \{0.2,1,1.2\} on 5 distillation-based continual learning methods iCaRL, Foster, Podnet, LUCIR, and R-DFCIL. It can be seen that our method is stable on the forgetting degree and accuracy despite a higher noisy interpretation, which is consistent with the conclusion of Theorem 1.
>
> **A6.** The unlabeled data is part of the memory buffer, with the remaining data being labeled. PsHD loss is applied only to the unlabeled data, while labeled data is distilled through the traditional KD. Thus, our method is compatible with the traditional KD.
>
> **Necessity of replaying both labeled and unlabeled data.** Labeled data provides accurate representations for discrimination, while unlabeled data has the potential to represent the entire dataset. Traditional knowledge distillation strategies are usually efficient in labeled data. Our PsHD loss is applied only to the unlabeled data, reserving intrinsic information and reducing computation costs, as illustrated in Answer 2.
>
> **A7.** The semi-supervised loss (SSL) used in our method follows the approach from DSGD, which is based on the SSL method Fixmatch. The SSL loss is not the primary motivation for our PsHD, as its effects in SSCL have already been explored in DSGD. We will clarify this point in the  Section 3.2.
>
> **A8-10.** The simplicial complex corresponding to the diagrams in Figure 2 is simplified, which leads to persistence feature inconsistency. We will update this figure to ensure greater consistency. We will correct the two typo issues.

---

> > ### Comment · Reviewer_K7Tp · 2024-08-12
> > **Answer**
> >
> > I appreciate the detailed answer. Authors did additional experiments to address my questions. Both clarifications and experiments must be included into manuscript. After some consideration, I'm raising my score. While I don't see evident mistakes in the paper, the final score is defined by overall novelty/impact of the paper.

---

> > > ### Author Response · Authors · 2024-08-12
> > >
> > > We sincerely appreciate the reviewer for fast reply. Meanwhile, we are very grateful for the reviewer's recognition of our responses and work, while increasing the final rating to ‘borderline accept’. Great thanks again!

---

### Official Review · Reviewer_AVV3 · 2024-07-18

**Soundness:** 3
**Presentation:** 4
**Contribution:** 3
**Rating:** 5
**Confidence:** 5

**Summary:**

This paper proposed to preserve intrinsic structural information with the use of persistent homology, so as to improve knowledge distillation and memory replay in semi-supervised continual learning. The authors provided an efficient acceleration algorithm to reduce computational overheads and theoretically demonstrated its stability. Extensive experiments demonstrate the effectiveness of the proposed method.

**Strengths:**

1. This paper is well organized and easy to follow. The motivation is clearly presented. Most of the related work has been discussed.

2. The design of persistent homology for knowledge distillation is reasonable to me. Although it seems to be an incremental design, the authors have provided many adaptations, such as the acceleration algorithm and the theoretical analysis.

3. The experimental validation is essentially extensive. It covers different setups of semi-supervised continual learning, ablation study, visualization, etc.

**Weaknesses:**

1. Although the authors have provided extensive experiments, the effectiveness of the proposed method seems to be limited. It offers less than 1% improvements in a majority of cases with relatively simple datasets (e.g., CIFAR-10 and CIFAR-100). This may be due to the classic NP-hard problem in selecting a few data to represent the entire dataset.

2. Although the authors have provided an acceleration algorithm to reduce computational overheads and conceptually analyzed it with their own method, I suggest to compare the overall resource overheads (storage and computation) of their method with other strong baselines (e.g., NNCSL, DER_Fix, and DSGD).

**Questions:**

I think this is essentially a solid paper. My major concerns lie in the effectiveness and efficiency. Please refer to the Weaknesses.

**Limitations:**

In Checklist the authors claimed that they have discussed the limitations in Section 5. However, Section 5 only presents the Conclusion without any discussion of the limitations.

---

> ### Author Rebuttal · Authors · 2024-08-07
>
> **A1**. The limited replied 500 samples restrict the improvement space, while our method reduces the degree of forgetting (BWT) by a substantial and balanced margin compared to the newly strong methods. Furthermore, our methods demonstrate effective utilization of replayed unlabeled samples, as evidenced by the substantial improvements over strong baselines when the memory buffer size is increased.
>
> **(1) Lower forgetting degree with comparable incremental accuracy.** As shown in Table 1 of the attached PDF file, our method surpasses the NNCSL by 6.15\% with a decreased average BWT of 18\%, outperforms DER\_Fix by 1.63\% with a decreased average BWT of 6.52\% forgetting on average, and exceeds DSGD by 0.89\% with a decreased average BWT rate 3.35\% in the four benchmarks.
>
> **(2) More balanced forgetting degree.** The BWT our method is under 27 in these benchmarks, avoiding the high forgetting rate observed with NNCSL. The overall BWT is lower than state-of-the-art with even higher incremental accuracy. These results show that our method PsHD achieves a better trade-off between adapting old tasks and preserving new tasks.
>
> **(3) Sufficient utilization of replayed unlabeled samples.** As shown in Table 1, when the number of replayed sampled increases to 5120, our method leverages the topology structure information of the unlabeled data more effectively, surpassing NNCSL and DSGD by 3.28\% and 2.07\% on average across the four benchmarks, respectively.
>
> **Table 1. Comparison on CIFAR10 and CIFAR100 with 5120 examples replayed**
> | Methods | CIFAR10_5% | CIFAR10_25% | CIFAR100_5% | CIFAR100_25% |
> |-------|------------|-------------|-------------|--------------|
> | Ours  | 82.5       | 87.2        | 47.5        | 58.6         |
> | NNCSL | 79.3       | 81.0        | 46.0        | 56.4         |
> | Compare_Ours | **+3.2**   | **+6.2**        | **+1.5**        | **+2.2**         |
> | DER_FIX | 80.3      | 78.2        | 43.6        | 56.3         |
> | DSGD  | 81.2       | 84.4        | 44.6        | 57.3         |
> | Compare_Ours | **+1.3**  | **+2.8**       | **+2.9**       | **+1.3**        |
>
> **A2.** Our method exhibits better performance, fewer memory budgets, and lower computation costs compared to these strong baselines. We are sorry for not fully expressing these advantages. The detailed verification is illustrated as follows.
>
> **(1) Highest accuracy with aligned memory size.** To verify the effectiveness and efficiency of our PsHD, we provide a comprehensive comparison of the overall resource overheads in Table 2, including the computation time and storage costs. The experiments are conducted on the same server for a fair comparison. Referring to the storage costs with aligned example 5120, our method achieves the highest accuracy at 47.5\% with the aligned memory size of 30.8 MB. It also surpasses the NNCSL by 1.5\% while using 45\% less memory, due to our smaller model buffer.
>
> **(2) Optimal solution in terms of both effectiveness and efficiency.** When reducing the example size to 2000, our method still achieves the best performance 46.0\% while reducing training time by 35.2\%, examples buffer size by 60.9\%, and memory size by 28.4\% compared to the best indices, which do not belong to the same method. Therefore, while existing strong baselines excel in different aspects —NNCSL in accuracy, DER\_Fix in training time, and DSGD is relatively balanced computation costs and accuracy, our method achieves the overall state-of-the-art performance with the least resource overheads and the highest accuracy.
>
>
> **Table 2. Effectiveness and efficiency comparison on CIFAR100_5%. The memory size is the sum of the storage of model parameters and replayed examples.**
>
> | Methods | Training time(h) | Parameters(m) | Example | Memory size (MB) | Acc(\%) |
> |-------|-----------|---------------|---------|------------------|---------|
> | NNCSL | 67        | 11.8          | 5120    | 56.58            | 46.01    |
> | DER_Fix | 34.1    | 4.6           | 5120    | 30.80            | 43.61   |
> | DSGD  | 40.8      | 4.6           | 5120    | 30.80            | 44.61   |
> | PsHD  | 40.4      | 4.6           | 5120    | 30.80            | **47.50** |
> | PsHD  | **22.1**  | 4.6           | **2000**| **22.07** |     _46.02_  |
>
>
> We thank the reviewer for the insightful suggestion to highlight the efficiency comparison,  and we have reorganized our expression  to highlight the superiority of the proposed method.
>
> **A3.** One limitation of the proposed method is that the partial computation of persistence homology is conducted on the CPU, which increases training time. We have include this limitation in the Conclusion.
>
> **Further improvement plan on limitation.** Although our acceleration algorithm mitigates this issue without disrupting end-to-end training, there is still room for improvement. The primary reason for this limitation is that the Python package Gudhi has not been developed for GPU use. To address this problem, we plan to implement persistence homology computation on the GPU, which is expected to further reduce training time.

---

> > ### Comment · Reviewer_AVV3 · 2024-08-11
> >
> > I thank the authors for their rebuttal. After considering the rebuttal and other reviewers' comment, I keep my score unchanged.

---

> > > ### Author Response · Authors · 2024-08-11
> > >
> > > We sincerely thank the reviewer for very fast reply. Meanwhile, we are very grateful that the reviewer recognizes our responses and work. Great thanks again!

---

### Author Rebuttal · Authors · 2024-08-07

We thank the reviewers for recognizing novelty (AVV3, K7Tp, 2waA), well-organization (AVV3, K7Tp), significance and reproducibility (AVV3,  K7Tp, 2waA), good performance (AVV3, K7Tp, 2waA), and comprehensive comparisons (AVV3, K7Tp, 2waA) of proposed PsHD.

**Reviewer AVV3's Questions and Our Responses:**

(1) For the question of  limited performance in Table 1, we providd additional comparison on the degree of forgetting (BWT) in Table 1 of attached PDF file, validating the superiority of proposed method on overcoming catastrophic forgetting.

(2) For the question of effectiveness and efficiency demonstration, we supplement detailed verification results in Table 2 of AVV3 rebuttal area.

(3) For the question of limitation clarficaiton, we clarified the CPU computation limitation and propose a further improvement strategy.

**Reviewer K7Tp's Questions and Our Responses:**

(1) For  the results of  traditional KD [1] and its varients from [2] employed on SSCL, we provide the evidence of negative influence of traditional KD [1] on SSCL in Table 2 of attached PDF file, and include experiment comparison of representative knowledge distillation strateties in [2] on the same benchmark, with results shown in Table 3 of the attached PDF file.


(2) For the missed comparison of reference [3], we acknolwedge the issue since its the same period work. We also conduct comparison experiment on four benchmarks in Table 3 of attached PDF file.

(3) For the stability evaluation of our methods, we add Gaussian noise interference experiments in Table 4 of attached PDF file.

(4) For more specific technique questions, such as how differentiate  PsHD, we also clarify the details in the K7Tp rebuttal area.

**Reviewer 2waA 's Questions and Our Responses:**

(1) For the unsufficient evidence for supporting the superiority, we supplement more experiments results to valid the conclusion from Table 1-4 of attached PDF file.

(2) For the unclear presentation of Figure 4, we conducte additional ablation experiments on larger distillation loss weight $\lambda$ and  add legends on each diagram for readability, the improved figure is depicted in Figure 1 of  attached PDF file.

(3) For the language problem, we improve the modifier of experiment results and illustrate the limitations at the same time. Such as use 'comparable' or 'competitive' to illustrate the accuracy improvement less than 1\%.

Reference

[1] Hinton, G., Vinyals, O., & Dean, J. (2015). Distilling the knowledge in a neural network. arXiv preprint arXiv:1503.02531.

[2] Li, S., Su, T., Zhang, X., & Wang, Z. (2024). Continual Learning with Knowledge Distillation: A Survey. Authorea Preprints.

[3] Kim, J., You, J., Lee, D., Kim, H. Y., & Jung, J. H. Do Topological Characteristics Help in Knowledge Distillation?. In Forty-first International Conference on Machine Learning.

---

### Author Response · Authors · 2024-08-12

Dear AC and reviewers,

We greatly thank for your time and effort to review our paper, while giving many valuable suggestions. As a kind reminder, we provided the detailed responses to the comments. Particularly, we sincerely thank the Reviewer AVV3 for very fast reply and recognition. We know and understand the reviewers are very busy, and sincerely hope Reviewer K7Tp and Reviewer 2waA could take a little time to review our responses, while let us know whether you have further questions. We are willing to try our best to answer these questions. Great thanks!

Best wishes

---

### Decision · Program_Chairs · 2024-09-25

**Decision:**

Accept (poster)

**Comment:**

The paper leverages persistent homology to enhance knowledge distillation and memory replay in semi-supervised continual learning, introducing an efficient algorithm to reduce overhead and demonstrating its stability and effectiveness through experiments. However, the improvement is quite incremental in many cases.

This paper is borderline for acceptance. After the rebuttal, all reviewers have given a score of 5. The previously raised issues regarding experiments and unclear concepts have been adequately addressed. To make a final decision, I carefully reviewed the paper, initial reviews, rebuttal, and all discussions. My assessment aligns with the reviewers' opinions: after revisions, the paper does not contain any obvious errors or major weaknesses, but the overall experimental impact and contribution to the field remain limited.

I am currently leaning toward acceptance but will discuss further with the SAC.